# Hyaluronan-Mediated Motility Receptor (HMMR) Overexpression Is Correlated with Poor Survival in Patients with B-ALL

**DOI:** 10.3390/ijms26020744

**Published:** 2025-01-16

**Authors:** Josselen Carina Ramírez-Chiquito, Vanessa Villegas-Ruíz, Isabel Medina-Vera, Itzel Sánchez-Cruz, Christian Lizette Frías-Soria, Marcela Concepción Caballero Palacios, Gabriela Antonio-Andrés, Alejandra Elizabeth Rubio-Portillo, Liliana Velasco-Hidalgo, Mario Perezpeña-Diazconti, Cesar Alejandro Galván-Diaz, Norma Candelaria López-Santiago, Sara Huerta-Yepez, Sergio Juárez-Méndez

**Affiliations:** 1Experimental Oncology Laboratory, National Institute of Pediatrics, Mexico City 04530, Mexico; carina-rc@outlook.com (J.C.R.-C.); vanessavillegasruiz@yahoo.com.mx (V.V.-R.); itzeuchan@hotmail.com (I.S.-C.); alejandra.rubio.portillo@gmail.com (A.E.R.-P.); 2Postgraduate in Biological Sciences, Postgraduate Unit, Building D, 1st Floor, Postgraduate Circuit, University City, Coyoacán, Mexico City 04510, Mexico; 3Research Methodology Department, National Institute of Pediatrics, Mexico City 04530, Mexico; isabelj.medinav@gmail.com; 4Molecular Pathology Laboratory, Department of Pathology, National Institute of Pediatrics, Mexico City 04530, Mexico; christian.frias@cinvestav.mx (C.L.F.-S.); mpdiazconti@gmail.com (M.P.-D.); 5Department of Pediatric Oncology, National Institute of Pediatrics, Mexico City 04530, Mexico; doc.marce1602@gmail.com (M.C.C.P.); lilianavh@hotmail.com (L.V.-H.); cesargalvan@mac.com (C.A.G.-D.); 6Oncology Research Unit, Hospital Infantil de México, Federico Gómez, Mexico City 06720, Mexico; gabya_24@yahoo.com.mx (G.A.-A.); shuertay@gmail.com (S.H.-Y.); 7Department of Pathology, National Institute of Pediatrics, Mexico City 04530, Mexico; 8Department of Hematology, National Institute of Pediatrics, Mexico City 04530, Mexico; nolsa99@yahoo.com

**Keywords:** leukemia, B-ALL, prognostic molecular marker, HMMR, RHAMM

## Abstract

Acute lymphoblastic leukemia (ALL) is a malignant neoplasm with the highest incidence in the pediatric population. Although the 5-year overall survival is greater than 85%, in emerging countries such as Mexico, the mortality rate is high. In Mexico, B-ALL is the most common type of childhood cancer; different characteristics suggest the presence of the disease; however, the prognosis is dependent on clinical and laboratory features, and no adverse prognostic molecular marker for B-ALL has yet been identified. The present research aimed to identify the prognostic value of HMMR expression in pediatric patients with B-ALL. The differential expression profile of B-ALL cells was determined via in silico analysis, and HMMR expression was subsequently measured via qRT–PCR and immunocytochemistry. The results were statistically analyzed via the ROUT test, Kolmogorov–Smirnov Z test, and Mann–Whitney U test. ROC curves and the Youden index were constructed, and Kaplan–Meier curves were plotted. We found that HMMR expression was increased in B-ALL patients (*p* < 0.0001). We observed that high expression was related to poor prognosis (*p* < 0.05). We observed that high expression was related to poor prognosis (*p* < 0.05). The increase in HMMR expression could be a potential early molecular prognostic marker and/or a new target in childhood B-ALL patients.

## 1. Introduction

Childhood cancer is one of the leading causes of death in children <15 years of age worldwide [1,2]. Leukemia is the childhood neoplasm most common in Mexico, and B-cell lymphoblastic leukemia (B-ALL) represents approximately 80% of the total cases. A diagnosis for B-ALL can be established if there is the presence of at least 20% blasts in the bone marrow or peripheral blood, and the risk stratification and disease prognosis are evaluated through cell morphology, immunophenotype, genetics, and cytogenetics, as well as clinical features such as age, sex, cell count at diagnosis, and the presence of blasts in the central nervous system and other extramedullary sites [3,4].

Although, in developed countries, survival after 5 years is promising (>90%), in developing regions, the likelihood of survival is reduced [4,5]. Several factors influence the low survival rate of patients with acute lymphoblastic leukemia, including socioeconomic status, population genetic factors, late diagnosis, risk-based stratification, and minimal residual disease monitoring [5,6].

In Mexico, more than 50% of patients with B-ALL are classified as high risk; unfortunately, almost 35% of all patients with B-ALL experience relapses, and fewer than 65% of all patients survive for 5 years, in contrast with other countries, such as Australia, the USA, and Japan, where the survival rate is more than 90% [6,7]. Furthermore, 55% of patients classified as standard risk has disease recurrence. Therefore, there is a need to improve molecular stratification for Mexican patients with B-ALL [6].

Numerous molecular alterations are known to be associated with ALL, including translocations (ETV6-RUNX1, TCF3-PXB1, and BCRA-ABL1), deletions, mutations, and amplifications of genes involved in histone modification (CREBBP, WHSC1, SETD2, EZH2, and TBL1XR1), transcription factors (PAX4, ERG, EBF1, and IZF1-3), tumor suppressors (RB1, CDKN2A/B, and TP53), kinases (PDGFRB, ABL1, JAK1/2, and EPOR), immune response receptors (CRFL2 and IL7R), genes related to the maturation of B cells (KMT2A, ERG, and SH2B3), apoptosis (BTG1), and the RTK–RAS pathway (FLT3, NRAS, and PTPN11) [4,8,9,10,11,12,13,14]. However, none of them have been classified as early diagnostic, prognostic, or relapse biomarkers in patients with ALL.

The hyaluronan-mediated motility receptor (HMMR), also known as RHAMM, CD168, or IHABP, is encoded by a gene located on chromosome 5, region q34, in the sense strand. This protein consists of 725 aa (84.2 kDa) and is a member of the hyaladherin family [15]; it contains two HA-binding domains (Bx7B) in the carboxyl terminus and lacks a transmembrane domain and peptide signal [16,17,18]. The cellular distribution of HMMR is variable; HMMR is localized to the membrane surface, cytoplasm, and nucleus and is involved in different cellular processes, such as motility, cell cycle regulation, and extracellular matrix degradation [19,20].

On the membrane surface, HMMR interacts with hyaluronan (HA), CD44, and growth factor receptors (TGFβ, bFRGFR, and PDGFR); this complex promotes cellular migration via the ERK1/2/MAPK signaling pathway [21,22,23]. Intracellularly, HMMR is attached to different proteins, such as actin filaments and microtubules, and modulates the dynamics of the cytoskeleton and centrosome [24,25]. In the nucleus, HMMR activates the kinases ERK1/2 and MEK and assembly factors of the mitotic spindle, such as TPX2, AURKA, and dynein [26]; these interactions switch on cascade signals to promote motility, cell cycle progression, and the transcription of genes related to invasion, such as MMP9 [27].

Recent studies have revealed that HMMR expression plays a crucial role in proliferation and metastasis in different types of cancer such as prostate [28], breast [29], and head and neck carcinomas [30]. In addition, findings from pancancer bioinformatics analysis have allowed the identification of HMMR overexpression in various tumors compared with healthy tissue and its association with lower survival [31]. However, studies of HMMR expression in pediatric cancer are limited. One study revealed that HMMR is associated with leukemic cells from patients with acute myeloid leukemia [32]. In another study, HMMR protein expression in paraffin blocks of bone marrow biopsies was associated with a poor prognosis in acute leukemia patients [33]. Since B-ALL represents 85% of acute lymphoblastic leukemia cases, determining the levels of HMMR mRNA and protein expression in bone marrow and peripheral blood samples from children with B-ALL is important. Therefore, the aim of this work was to determine HMMR expression-based prognoses in patients with B-ALL.

## 2. Results

### 2.1. HMMR Expression in B-ALL

The microarray data mining included four healthy controls and ten B-ALL patients whose data were analyzed via Affymetrix GeneChip 1.0. Bioinformatics analysis of the microarray gene expression data revealed that 1938 genes were dysregulated in B-ALL, among which 1102 genes were upregulated and 836 genes were downregulated (Figure 1A), with a fold change threshold of >2 or <−2 and a *p* value of <0.05. Among the differential expression profiles, HMMR was found to be overexpressed in B-ALL, with a fold change of 2.15 and a *p* value = 9.38^−14^ (Figure 1B). 

Next, we evaluated HMMR expression via quantitative RT–PCR in the bone marrow aspirates of 66 patients whose diagnosis was confirmed to be B-ALL and who were free of treatment at the time of biopsy. Our results revealed wide variability in HMMR expression levels (Figure 2A). Nevertheless, when patient survival was separated from death, we observed a significant increase in HMMR expression in patients who died (*p* value = 0.032) (Figure 2B). Additionally, the opposite behavior was detected in patients with relapse, where we observed a tendency toward HMMR suppression in the group with relapse; however, this result was not statistically significant, possibly because of the sample size (Figure 2C).

### 2.2. High HMMR Expression Predicts Death in Patients with B-ALL

To determine the impact of HMMR expression on the mortality of B-ALL patients, we constructed an ROC curve, and the analysis of the data revealed an AUC of 0.67 and a confidence interval of 95%. Subsequently, threshold values were calculated for HMMR expression with the Youden index, and we determined a cutoff value of 0.25 (Figure 3A). Finally, we constructed a Kaplan–Meier curve, where we observed that patients with the highest expression had a greater probability of death than patients with low expression (*p* = 0.02), with a hazard ratio of 5.1 (Figure 3B). These results suggest that the HMMR could be used as a test for the prognosis of patients with B-ALL because it has a sensitivity of 0.69 and a specificity of 0.74. Similarly, we analyzed HMMR expression data in B-ALL patients considering the presence of relapse, type of relapse (early or late), and risk classification (standard or high); however, the results were not statistically significant.

### 2.3. HMMR Is Overexpressed in the Peripheral Blood of Patients with B-ALL

Since HMMR mRNA expression could be a tool for the early prognosis of B-ALL patients, we evaluated HMMR expression at the protein level in the peripheral blood of 54 control samples (Figure 4A) and 63 B-ALL patients (Figure 4B). This analysis revealed that HMMR expression was significantly increased in B-ALL patients (*p* < 0.0001) (Figure 4C). 

When HMMR protein expression was associated with mortality and relapse characteristics, the data revealed that HMMR was overexpressed in patients who died from B-ALL (*p* value = 0.001) but not in those who relapsed (Figure 5A,B), which is consistent with what was observed at the bone marrow mRNA level (Figure 2B,C).

Similarly, the threshold values were established considering the number of cells positive for HMMR through the Youden index, such that >82 positive cells were considered to have high expression, whereas low expression corresponded to <82 positive cells. A Kaplan–Meier plot was constructed on the basis of the number of positive cells, as expected, and in accordance with previous results, a relationship between mortality and the number of HMMR-positive cells was observed (*p* = 0.027), with a risk factor of 4.9 (Figure 6B).

## 3. Discussion

Cancer is one of most important health problems in the world; in 2020, the World Health Organization reported 1.3 million new cancer cases [34]. In Mexico, leukemia is the most common pediatric malignant neoplasm, accounting for approximately half of all childhood neoplasms [35]. Acute lymphoblastic leukemia accounts for approximately 80% of all leukemias in children and adolescents under 14 years of age. In developing countries, approximately 90% of patients have a 5-year survival [36,37,38]. However, in Mexico, the survival rate is ~70% [39], possibly due to diagnosis at an advanced stage and other factors, such as cytogenetic, immunophenotype, and molecular features [40,41,42,43]. Although several clinical parameters have been clearly established to classify risk groups and treatments, more than half of standard-risk patients relapse during treatment, worsening their prognosis [7]. This could suggest that other characteristics have not been elucidated thus far [6]. Additionally, disease monitoring during treatment, immunotherapy, and cellular therapy is limited in low- and middle-income countries, such as ours.

Gene expression is a spectacular cellular process that has been shown to be tissue-dependent and can be modulated by the microenvironment. RNA is expressed as a molecular marker in several diseases, such as Mendelian diseases, tuberculosis, and some cancers [44,45,46]. However, quantification is not always homogenous, as it depends on the cellular condition, stress, temperature, and CO_2_ concentration, among other factors. Additionally, the cancer cells expressed several deregulated genes; in our case, approximately two thousand deregulated transcripts were observed (Figure 1A). However, it is difficult to validate a deregulated gene expression profile; in our case, we evaluated the random expression of deregulated genes, including ZNF695 [47], CENPE [48], and MYB [49]. Another significantly deregulated gene in B-ALL was HMMR, which was significantly overexpressed via gene expression profile analysis (Figure 1A) and validated via quantitative RT–PCR and immunocytochemistry (Figure 2 and Figure 4B). Under healthy conditions, HMMR expression is low, with the exception of in the testis and peripheral blood [22,27,50]. Our results are in accordance with those of previous reports [51,52]. Although HMMR is expressed in the peripheral blood of healthy controls, it is not comparable with its expression in leukemia patients, as we have observed in our findings (Figure 4C), which suggests that this gene is deregulated in B-ALL; however, we do not know which cells express HMMR in healthy peripheral blood because we do not perform cell sorting, and this will be a topic of future research.

In different cancer types, HMMR is overexpressed, including acute myeloid leukemia and ovarian, pancreatic, kidney, and liver cancers [53,54,55,56,57]. Our results show that HMMR is also overexpressed in pediatric B-ALL patients. Furthermore, HMMR expression has been observed in tumor stem cells, suggesting an important role in carcinogenesis [58]; for example, in myeloma, elevated HMMR expression is associated with cytogenetic abnormalities [59]. Moreover, in both endometrial tumors and non-small-cell lung cancer, high HMMR expression is correlated with a high tumor grade [60,61], and in bladder cancer, HMMR overexpression has been observed in areas of invasion, as well as in metastatic lung and colon tumors [61,62,63].

Elevated HMMR expression has also been reported to be correlated with poor prognosis in patients with tumors of the stomach, hepatocarcinoma, and B-cell chronic lymphocytic leukemia in adults [52,64,65], among others. Our results also revealed that high HMMR mRNA and protein expression is associated with an adverse prognosis in B-ALL pediatric patients (Figure 3B and Figure 6B), which is in agreement with the findings of Tarullo, Beck, and Tzankov in breast cancer, lymphoma, and acute myeloid leukemia, respectively [66,67,68].

In hematological malignancies such as myeloma, HMMR has been shown to correlate with structural abnormalities of centrosomes contributing to genomic instability, whereas in leukemias, HMMR can activate humoral and cellular immune responses; however, the biological role of HMMR in this type of neoplasm has not been fully elucidated, since HMMR may play different roles at both the intracellular and extracellular levels [20,32,69,70]. On the membrane surface, HMMR can promote cellular motility via the ERK1/2/MAPK cascade as well as in the cytoplasm, where it can also target actin filaments and microtubules to regulate cytoskeleton and centrosome dynamics [21,23,24,25,27]. In the nucleus, HMMR promotes cell cycle progression and the transcription of genes associated with the degradation of the extracellular matrix, such as MMP9 [26,27]. Therefore, the overregulation of any of these functions—such as sustained proliferation, invasion, and metastasis, events associated with the poor prognosis of patients with B-ALL—could be linked to cancer progression. These findings lead us to believe that HMMR acts at several cellular levels and that synergy may contribute significantly to poor prognosis. However, further research is needed on this topic.

On the other hand, even though the relationship between the initial number of blasts in the sample and HMMR expression was not studied in this analysis, Shalini S. et al. reported that there is no correlation between the concentration of blasts at diagnosis and the level of HMMR expression [33], which could indicate that the overexpression of HMMR observed at this cutoff in B-ALL patients is a result of the dysregulation of expression in leukemic cells and not of the healthy cells in the biopsy. 

Our results did not reveal a significant association between relapse and HMMR overexpression; however, we observed a tendency toward decreased expression of HMMR in patients with relapse. We did not observe significant differences, possibly due to the size of the current cohort that we analyzed, so it will be necessary to increase the number of patients to obtain a clearer picture of these findings. Additionally, some patients with high HMMR expression may develop resistance to chemotherapy, leading to death, similar to what has been observed in prostate and gastric cancers [71,72]. In addition, patients with low HMMR expression experience relapse.

Although these findings may seem contradictory, they may be because HMMR apparently acts at two levels. First, overexpression promotes exacerbated malignant activity [21], and second, underexpression promotes incorrect or weak roles [22], such as correct mitotic assembly [24,25]. This duality could be a consequence of the coexpression of HMMR messenger RNA variants, since transient colocalization of four HMMR variants has been observed [24].

## 4. Materials and Methods

### 4.1. Data Mining

For this investigation, the microarray data from GeneChip 1.0, Affymetrix, were obtained from free repositories, Gene Expression Omnibus (GEO) and ArrayExpress, according to previous reports [48,49]. The results that corresponded to patients with B-ALL and healthy B-cell individuals were included. Bioinformatic analyses were performed via .CEL files, and these raw data were processed via Partek Genomics Suite v6.6. The differentially expressed genes (DEGs) were selected on the basis of a *p* value < 0.005 and a fold change >2 and <−2 according to previous studies [73,74,75].

### 4.2. Biological Samples and Ethics Statement

In this study, 129 patients with a confirmed diagnosis of B-ALL were included: 66 samples from marrow bone for RNA extraction, 63 from peripheral blood for immunocytochemistry, and 54 from the peripheral blood of healthy donors; these patients were considered controls. All samples were collected after signed informed consent was obtained from the patients, and the protocol was approved by the Institutional Ethics Committee (INP 060/2016) in accordance with the Declaration of Helsinki. Age, sex, and patient characteristics are shown in Table 1.

### 4.3. RNA Purification

Total RNA was purified from the bone marrow of 66 patients with B-ALL. The lymphocyte fraction was obtained via a density gradient via Lymphoprep (STEMCELL Technologies, Vancouver, BC, Canada) according to the established protocol, and the RNA was purified from bone marrow samples via an RNeasy Mini Kit (Qiagen, Valencia, CA, USA) following the manufacturer’s instructions according to previous reports [47,74,76]. Total RNA was quantified via a NanoDrop One Spectrophotometer (Thermo Scientific Waltham, Waltham, MA, USA).

### 4.4. cDNA Synthesis

After RNA quantification, total RNA was treated with DNase (Thermo Fisher Scientific, Waltham, MA, USA) to avoid DNA contamination. The reactions were incubated at 37 °C for 30 min, and then, 1 µL of 50 mM EDTA was added and incubated at 65 °C for 10 min. cDNA synthesis was subsequently performed via the commercial RevertAid Transcriptase Kit (Thermo Fisher Scientific, Waltham, MA, USA), which uses a 1 mM dNTP mixture, 100 pmol random hexamers, 10 U RiboLock RNAse Inhibitor, and 200 U RevertAid Reverse Transcriptase (Thermo Fisher Scientific, Waltham, MA, USA). The reactions were incubated at 25 °C for 10 min, 42 °C for 60 min, and 70 °C for 10 min. The cDNA samples were stored at −20 °C until use.

### 4.5. RT–PCR Amplification

cDNA synthesis was evaluated via RT–PCR via the housekeeping gene RPS18 with the following primers: Fw 5′-AATCCACGCCAGTACAAGATCCCA-3′ and reverse 5′-TTTCTTCTTGGACACACCCACGGT-3′. RT–PCR was performed with KAPA2G Fast HotStart Ready Mix (Kapa Biosystems, Wilmington, MA, USA). The temperature cycle was as follows: predenaturation to 95 °C for 3 min; 40 cycles of 95 °C for 15 s, 60 °C for 15 s, and 72 °C for 15 s; and a final extension at 72 °C for 5 min. The products were analyzed via electrophoresis in 1.5% agarose gels.

### 4.6. Quantitative HMMR Expression

The HMMR of the mRNAs was determined via real-time PCR with specific primers (10 μM) (Fw 5′- GCGTTAACAGCCAGT GAGATAG-3′, Rv 5′-TGCTGAACATCCTCTGCATTT-3′) via Sybr Fast qPCR Master Mix (2x) (Kapa Biosystems Inc., Wilmington, MA, USA), and the expression gene RPS18 (Fw 5′-CAGCCAGGTCCTAGCCAATG -3′, Rv 5′-CCATCTATGGGCCCGAATCT-3′) was used as a reference gene. Quantitative RT–PCR was performed on a Step One Real-Time PCR System (Applied Biosystems Inc., Foster City, CA, USA). The reactions were incubated as follows: 95 °C for 10 min; 40 cycles of 95 °C for 10 s and 60 °C for 30 s; and finally a melting curve at 51 °C for 1 min with quantification every 0.3 °C for 15 s up to 95 °C. Relative quantification analysis (method 2^−ΔΔCT^) was performed according to previous reports [47,49,77].

### 4.7. Sample Processing and Immunocytochemistry

Peripheral blood cells from 63 patients with B-ALL and 54 controls were collected from pediatric patients and processed to isolate mononuclear cells (PBMCs). Gradient separation was performed via Ficoll-Paque™ PLUS (GE Healthcare) following the manufacturer’s protocol. Blood samples were diluted with an equal volume of phosphate-buffered saline (PBS) before being overlaid with Ficoll-Paque™ PLUS at a 1:1 ratio. The samples were subsequently centrifuged at 400× *g* for 30 min at room temperature without braking. The mononuclear cell layer was carefully collected, washed twice with PBS to remove residual Ficoll, and resuspended in PBS.

The cell suspensions were then adjusted to a concentration of 1 × 10^4^ cells per slide. The cells underwent a cytospin and were placed onto glass slides, fixed with 4% paraformaldehyde for 15 min at room temperature, and stored at 4 °C until immunostaining according to previous studies [78,79].

Immunostaining was performed on the prepared cell slides to assess specific markers. To minimize inter-assay variability, all the slides were immunostained in a single batch. The process began with antigen retrieval via 0.01 M sodium citrate buffer (pH 6.0). Endogenous peroxidase activity was quenched with methanol and hydrogen peroxide. Nonspecific binding was blocked with 2% normal swine serum. Then, the primary rabbit monoclonal antibody CD168/RHAMM [EPR4055] GTX62573 (GeneTex, Irvine, CA, USA) was added at a dilution of 1:1500, and the slides were incubated overnight at 4 °C in a humid chamber with gentle agitation. Following washing, the slides were treated with a biotin-conjugated secondary antibody and subsequently with HRP-conjugated streptavidin (Universal LSAB + KIT/HRP, VECTOR Laboratories). Diaminobenzidine (DAB) was used as the chromogen, and hematoxylin was applied for counterstaining. After dehydration, the slides were mounted with resin. Rabbit immunoglobulin G (Sc-2027) (Santa Cruz Biotechnology, Dallas, TX, USA) was used as an isotype control [80,81].

The slides were analyzed via an Olympus BX-40 microscope with brightfield optics. Positively stained cells, identified by their brown color (DAB staining), were quantified in four randomly selected fields per slide. Each field was evaluated via Image-Pro Plus^®^ image analysis software v6.2 (Media Cybernetics, Rockville, MD, USA).

For each field, the total number of cells and the number of HMMR-positive cells were counted. The percentage of positive cells (% positivity) was calculated for each slide by dividing the number of positively stained cells by the total number of cells and multiplying by 100. This calculation was performed for all four fields, and an average percentage of positive cells was reported for each slide, providing a quantitative measure of immunostaining efficiency and consistency [80,81].

### 4.8. Statistical Analysis

The data were treated to identify outliers via the ROUT method (Q = 1%), and the variables were evaluated via the Kolmogorov–Smirnov Z test to examine the distribution type. The data are expressed as the median (range (25th percentile at 75th percentile)), and the differences between patients with adverse prognoses and those without adverse prognoses were analyzed via the Mann–Whitney U test, with *p* < 0.05 considered significant. ROC curves were constructed, and the area under the curve (AUC) and the 95% confidence interval (95% CI) were subsequently calculated.

The cutoff points for the expressed genes were determined through the Youden index (Y = sensitivity + specificity − 1), taking into account values closer to 1. Finally, Kaplan–Meier curves were used to represent survival time, and hazard ratios (HRs) were calculated via Cox proportional risk analysis. We considered *p* < 0.05 to indicate statistical significance. The data were analyzed via GraphPad Prism version 9.0.2 software for Windows (Boston, MA, USA).

## 5. Conclusions

High HMMR expression may serve as an early molecular marker for poor prognosis in patients with B-ALL, indicating a potential role in identifying high-risk individuals. Its overexpression could help stratify patients for more aggressive treatment protocols and highlight the need for targeted therapeutic approaches. Further studies are needed to validate the HMMR as a prognostic biomarker and explore its potential as a novel therapeutic target in pediatric B-ALL patients.

## Figures and Tables

**Figure 1 ijms-26-00744-f001:**
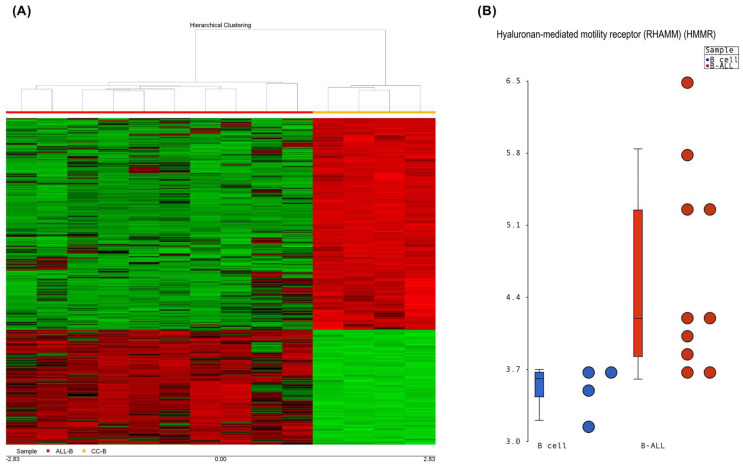
Gene expression profile associated with B-ALL. (**A**) Heatmap illustrating the DEGs in B-ALL patients, with an FC >2 or <−2 and a *p* value <0.05. Red indicates decreased expression, and green indicates increased expression. (**B**) Bioinformatic analysis of the expression microarray revealed that HMMR is overexpressed in patients with B-ALL (orange dots) compared with normal B cells (blue dots), with an FC of 2.15 and a *p* value = 9.38^−14^.

**Figure 2 ijms-26-00744-f002:**
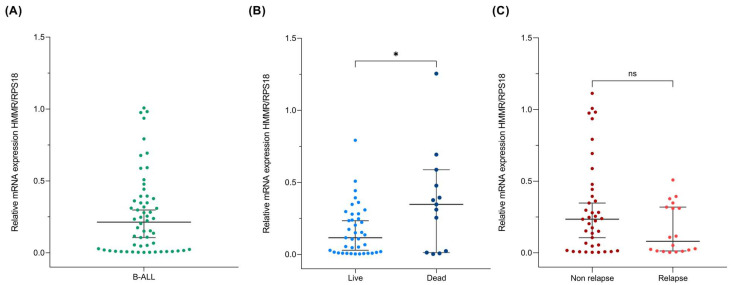
High HMMR relative expression is associated with mortality in B-ALL patients. (**A**) Dot plot showing the HMMR relative quantification in the bone marrow samples of patients with B-ALL. (**B**) HMMR relative expression in living and deceased patients. The data revealed that HMMR was overexpressed in patients who died, with a *p* value of 0.032. (**C**) HMMR relative expression in B-ALL patients with and without relapse; the data tend to be suppressed in patients with disease recurrence. * = statistical significance (*p* < 0.03), ns = non statistical significance.

**Figure 3 ijms-26-00744-f003:**
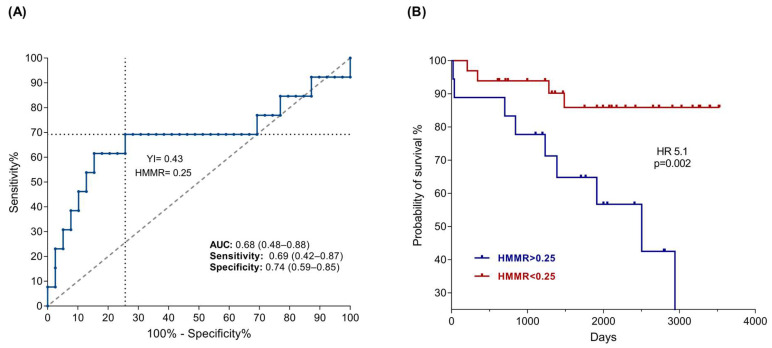
High expression of HMMR mRNA is associated with poor prognosis in B-ALL patients. (**A**) ROC curve for B-ALL mortality showing the values of the AUC, sensitivity, specificity, Youden index, and cutoff value, with a *p* value of 0.05. (**B**) Kaplan–Meier curve analysis revealed that high expression of HMMR was associated with shorter survival (red line) than low expression (blue line), with a *p* value < 0.05.

**Figure 4 ijms-26-00744-f004:**
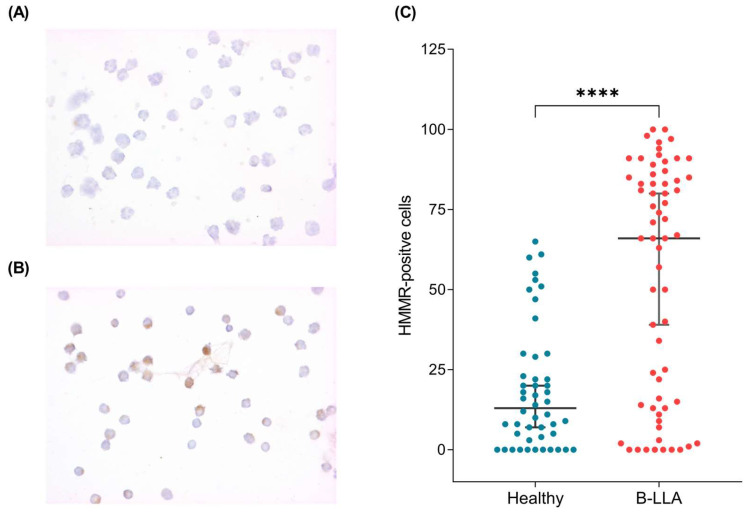
HMMR protein expression in healthy donors and patients with B-ALL. (**A**) Representative immunocytochemistry images of HMMRs (40×) in the peripheral blood of healthy donors and (**B**) B-ALL patients. In (**C**), the graphic represents HMMR protein expression in healthy donors (blue dots) and in patients with B-ALL (red dots), with significant differences (**** = *p* value < 0.0001).

**Figure 5 ijms-26-00744-f005:**
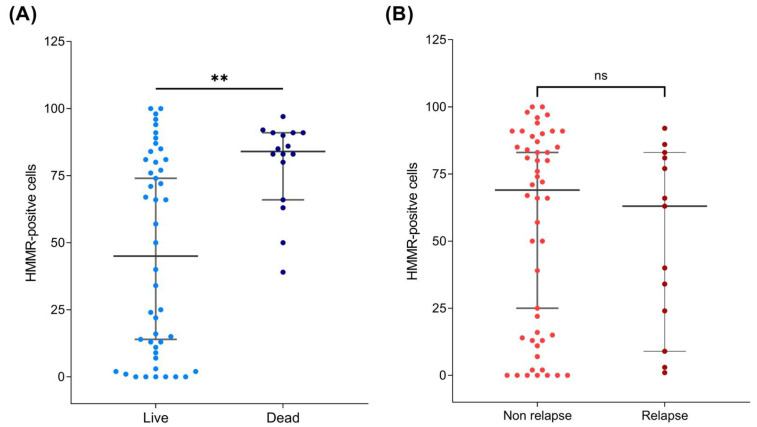
High HMMR protein expression is associated with mortality in B-ALL patients. Dot plot showing the number of HMMR-positive cells in the peripheral blood samples of patients with B-ALL. (**A**) HMMR protein expression in living and deceased patients. The data revealed that HMMR was overexpressed in patients who died, with a *p* value of 0.001. (**B**) In relapsed and nonrelapsed B-ALL patients, the data revealed no statistically significant changes in HMMR expression. ** = statistical significance (*p* < 0.002), ns = non statistical significance.

**Figure 6 ijms-26-00744-f006:**
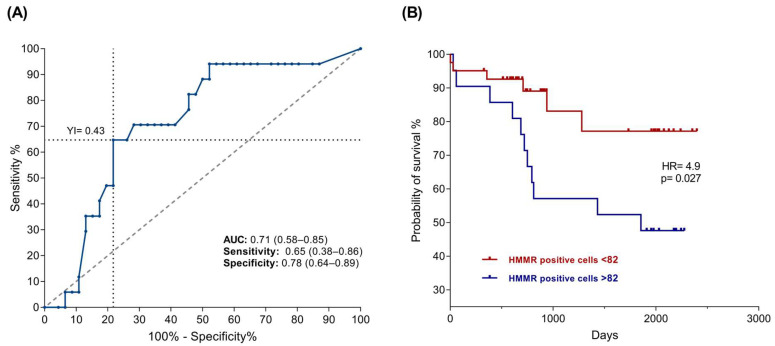
High HMMR protein expression is associated with poor prognoses in B-ALL patients. (**A**) ROC curve showing the values of the AUC, sensitivity, specificity, and Youden index, with a *p* value of 0.002. (**B**) Kaplan–Meier curve showing survival days in patients with B-ALL. The red line corresponds to high HMMR expression, and the blue line corresponds to low HMMR expression. The results revealed a statistically significant difference, with a *p* value < 0.05.

**Table 1 ijms-26-00744-t001:** Demographic and clinical characteristics of the included B-ALL patients and controls.

B-ALL Patients	N = 129 (100%)
**Gender**	
Female	65 (50.4%)
Male	64 (49.6%)
**Age**	
Age at diagnosis in years, mean ± SD	7.7 ± 4.5 years
Most common age at diagnosis	3 years (14.0%)
Age groups in years	
0–4	46 (35.7%)
5–8	30 (23.3%)
9–12	25 (19.4%)
13–18	28 (26.4%)
**Risk**	
Standard	30 (23.3%)
High	99 (76.7%)
**Relapse**	
Yes	34 (26.4%)
No	95 (73.6%)
**Alive**	97 (75.2%)
Alive with relapse	20 (15.5%)
Alive without relapse	77 (59.7%)
**Deceased**	32 (24.8%)
Deceased with relapse	14 (10.8%)
Deceased without relapse	18 (14.0%)
Survival days, mean ±SD	1562 ± 83
Disease-free survival days, mean ±SD	574 ± 93
**Controls**	N = 54 (100%)
**Gender**	
Female	25 (46.3%)
Male	29 (53.7%)
**Age**	
Mean age ± SD	11.6 ± 1.6 years

## Data Availability

The personal data of the patients are not available due to ethical reasons.

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
