# Peer review of "Hyaluronan-Mediated Motility Receptor (HMMR) Overexpression Is Correlated with Poor Survival in Patients with B-ALL"

_ijms, 2025, doi:10.3390/ijms26020744_

Round 1
Reviewer 1 Report
Comments and Suggestions for Authors
In this manuscript, the authors showed high expression of HMMR is a prognostic marker for B-ALL. They first mined the HMMR from public available microarray B-ALL data base. Then they tested HMMR mRNA level from patient sample and determined a cutoff point to stratify all the survival data from their 129 patient and depicted that high HMMR mRNA level is associated with shorter survival. Additionally, they examined the protein expression of HMMR through immunohistochemistry and showed high HMMR expression is an adverse marker for death but not relapse in B-ALL patients. The association between CD168/HMMR and pediatric leukemia has been reported repeatedly previously with similar methods (Shalini, Hematology, 2018; Giannopoulos, Blood, 2006). There is a lack of novelty of this manuscript. Additionally, the authors can improve this manuscript in the following ways:
1. The authors should include more data points in microarray data mining in Figure 1.
2. The authors should add more details about the immunohistochemistry method, especially the cell number dilution part, as higher blast number in B-ALL patients could lead to higher positive HMMR cells. And the authors should provide a frequency of HMMR-positive cells in each patient, better in another method, like flow-cytometry.
3. The authors should discuss why the HMMR expression is only associate with survival but not relapse.
4. There are some typos through the manuscript and some sentences are confusing: 1) the description in line 43 “preceded only by accidents” is confusing; 2) line 50 “in developed region” seems meaning “ in developing region”. 3) the sentence from line 58 to 60 is not clear - “ Although more than 50% of our population is classified as high risk, the frequency of relapse in patients stratified as standard risk is 55%, suggesting the need to improve stratification and implement sub-stratification for Mexican patients with B-ALL.” Please rephrase this sentence. 4) Line 178, “B-LLA” should be a typo.
Comments on the Quality of English LanguageIn this manuscript, the authors showed high expression of HMMR is a prognostic marker for B-ALL. They first mined the HMMR from public available microarray B-ALL data base. Then they tested HMMR mRNA level from patient sample and determined a cutoff point to stratify all the survival data from their 129 patient and depicted that high HMMR mRNA level is associated with shorter survival. Additionally, they examined the protein expression of HMMR through immunohistochemistry and showed high HMMR expression is an adverse marker for death but not relapse in B-ALL patients. The association between CD168/HMMR and pediatric leukemia has been reported repeatedly previously with similar methods (Shalini, Hematology, 2018; Giannopoulos, Blood, 2006). There is a lack of novelty of this manuscript. Additionally, the authors can improve this manuscript in the following ways:
1. The authors should include more data points in microarray data mining in Figure 1.
2. The authors should add more details about the immunohistochemistry method, especially the cell number dilution part, as higher blast number in B-ALL patients could lead to higher positive HMMR cells. And the authors should provide a frequency of HMMR-positive cells in each patient, better in another method, like flow-cytometry.
3. The authors should discuss why the HMMR expression is only associate with survival but not relapse.
4. There are some typos through the manuscript and some sentences are confusing: 1) the description in line 43 “preceded only by accidents” is confusing; 2) line 50 “in developed region” seems meaning “ in developing region”. 3) the sentence from line 58 to 60 is not clear - “ Although more than 50% of our population is classified as high risk, the frequency of relapse in patients stratified as standard risk is 55%, suggesting the need to improve stratification and implement sub-stratification for Mexican patients with B-ALL.” Please rephrase this sentence. 4) Line 178, “B-LLA” should be a typo.
Author Response
We are very grateful for your response and the opportunity to revise our manuscript: “Hyaluronan-mediated motility receptor (HMMR) overexpression is correlated with poor survival in patients with B-ALL” International Journal of Molecular Sciences.
We carefully considered the comments offered by both reviewers, which were highly insightful and enabled us to greatly improve the quality of our manuscript.
We want to extend our appreciation for taking the time and effort necessary to provide such insightful comments. We apologize for the linguistic quality of the original manuscript, which has been corrected.
Herein, we explain how we revised the paper on the basis of those comments and recommendations. Revisions in the text are shown with the marked-up track changes function from MS words. We hope that the revisions in the manuscript and our accompanying responses will be sufficient to make our manuscript suitable for publication in the International Journal of Molecular Sciences. Next, we detail our responses to each reviewer’s concerns and comments.
Reviewer 1
COMMENT: The association between CD168/HMMR and pediatric leukemia has been reported repeatedly previously with similar methods (Shalini, Hematology, 2018; Giannopoulos, Blood, 2006). There is a lack of novelty of this manuscript.
RESPONSE: Thank you for this valuable comment. The expression of CD168/HMMR has been evaluated in pediatric leukemia patients. However, the evaluation has been performed via immunohistochemistry (Shalini, Hematology, 2018) (Sai Shalini, Hematology, Transfusion and Cell Therapy, 2018). Additionally, they compared the different types of leukemias (Shalini, Hematology, 2018) and T-cell lymphoid leukemia (Giannopoulos, Blood, 2006) in our case. Weevaluated the expression of HMMR in B-cell lymphoid leukemia and evaluated its effects via immunohistochemistry and quantitative RT‒PCR in two independent cohorts with a clinical follow-up of more than 7 years.
COMMENT: The authors should include more data points in microarray data mining in Figure 1.
RESPONSE: We have included more information on differentially expressed genes in acute lymphoblastic leukemia in our expression analysis and added a heatmap resulting from that analysis.
COMMENT: The authors should add more details about the immunohistochemistry method, especially the cell number dilution part, as higher blast number in B-ALL patients could lead to higher positive HMMR cells. And the authors should provide a frequency of HMMR-positive cells in each patient, better in another method, like flow-cytometry.
RESPONSE: Compared with flow cytometry, immunocytochemistry (ICC) offers several advantages for evaluating protein expression in PBMCs. ICC allows precise visualization of protein localization within specific cellular compartments, providing spatial context that flow cytometry cannot offer. It enables simultaneous analysis of protein expression and cell morphology, making it possible to correlate protein presence with specific cellular states or abnormalities. Additionally, ICCs provide single-cell resolution and facilitate the direct observation of heterogeneity within the PBMC population, identifying variations in protein expression across cell types on the same slide. While variations in the proportion of blasts in patient samples may influence the percentage of positive cells, this can be addressed by carefully selecting representative fields for analysis and normalizing data on the basis of total cell counts per slide. CC is an excellent option for studies such as ours, where only small sample volumes are available owing to the pediatric nature of the patient population.
There are several studies in which PBMCs from patients with acute lymphoblastic leukemia (ALL) are used to evaluate the expression of various proteins via ICC, similar to our approach, which has yielded promising results. Furthermore, our findings are consistent with the data obtained via RT‒PCR in these same studies (Refs).
-Jean Hughes Dalle 1, Martine Fournier, Brigitte Nelken, Françoise Mazingue, Jean-Luc Laï, Francis Bauters, Pierre Fenaux, Bruno Quesnel. p16(INK4a) immunocytochemical analysis is an independent prognostic factor in childhood acute lymphoblastic leukemia. Blood. 2002 Apr 1;99(7):2620-3. doi: 10.1182/blood.v99.7.2620.
- M L Den Boer 1, C M Zwaan, R Pieters, K M Kazemier, M M Rottier, M J Flens, R J Scheper, A J Veerman. Optimal immunocytochemical and flow cytometric detection of P-gp, MRP and LRP in childhood acute lymphoblastic leukemia. Leukemia. 1997 Jul;11(7):1078-85. doi: 10.1038/sj.leu.2400729.
COMMENT: The authors should discuss why the HMMR expression is only associate with survival but not relapse.
RESPONSE: We have added information to the discussion concerning why we did not find significant differences in B-ALL relapse.
COMMENT: There are some typos through the manuscript and some sentences are confusing: 1) the description in line 43 “preceded only by accidents” is confusing; 2) line 50 “in developed region” seems meaning in developing region”. 3) the sentence from line 58 to 60 is not clear - “ Although more than 50% of our population is classified as high risk, the frequency of relapse in patients stratified as standard risk is 55%, suggesting the need to improve stratification and implement sub-stratification for Mexican patients with B-ALL.” Please rephrase this sentence. 4) Line 178, “B-LLA” should be a typo. “
RESPONSE: We have thoroughly revised the manuscript and have corrected the errors that you have kindly pointed out. In addition, we have eliminated redundant sentences and added missing information throughout the manuscript.

Reviewer 2 Report
Comments and Suggestions for Authors
The manuscript “Hyaluronan-mediated motility receptor (HMMR) overexpression is correlated with poor survival in patients with B-ALL by Ramírez-Chiquito, JC et al. asserts that elevated expression of the HMMR gene is associated with various cancers, including B-ALL, and is linked to poor prognosis. The authors report increased HMMR expression at both mRNA and protein levels in pediatric B-ALL, which correlates with adverse outcomes. This observation aligns with findings in other malignancies such as breast cancer, lymphoma, and acute myeloid leukemia.
The analyses conducted by the authors are relevant and contribute to the prediction of disease progression. However, the precise biological role of HMMR overexpression in pediatric B-ALL remains unclear. The authors should specify and discuss which future studies might focus on differentiating the intracellular versus extracellular mechanisms of HMMR that contribute to leukemogenesis.
In the Results section, it is crucial to clarify which blood cells are expressing HMMR and being quantified. This information would strengthen the interpretation of the data.
The authors should also propose potential approaches for targeting HMMR therapeutically and outline how these could yield translational insights for clinical applications.
Additionally, the manuscript should address differences in HMMR expression across various types of leukemia and other hematological malignancies, as this is necessary for inclusion in the Discussion section.
The authors should provide an explanation for the lack of statistically significant differences in HMMR expression between relapsed and non-relapsed B-ALL patients, as this discrepancy appears to be unrelated to the amount of HMMR expressed.
Finally, in Figure 5B, the Y-axis label is missing and should be added for clarity.
Author Response
We are very grateful for your response and the opportunity to revise our manuscript: “Hyaluronan-mediated motility receptor (HMMR) overexpression is correlated with poor survival in patients with B-ALL” International Journal of Molecular Sciences.
We carefully considered the comments offered by both reviewers, which were highly insightful and enabled us to greatly improve the quality of our manuscript.
We want to extend our appreciation for taking the time and effort necessary to provide such insightful comments. We apologize for the linguistic quality of the original manuscript, which has been corrected.
Herein, we explain how we revised the paper on the basis of those comments and recommendations. Revisions in the text are shown with the marked-up track changes function from MS words. We hope that the revisions in the manuscript and our accompanying responses will be sufficient to make our manuscript suitable for publication in the International Journal of Molecular Sciences. Next, we detail our responses to each reviewer’s concerns and comments.
Reviewer 2
COMMENT: The analyses conducted by the authors are relevant and contribute to the prediction of disease progression. However, the precise biological role of HMMR overexpression in pediatric B-ALL remains unclear. The authors should specify and discuss which future studies might focus on differentiating the intracellular versus extracellular mechanisms of HMMR that contribute to leukemogenesis.
RESPONSE: We fully agree with your observation that the biological role of HMMR expression in leukemia is not clear. However, we determined that its increase is significantly related to the death of patients. Because HMMR is located in different areas of the cell, it is even more difficult to pinpoint its function. We have added information to the discussion about the possible role of HMMR and its function in the relapse and death of patients with leukemia.
COMMENT: In the Results section, it is crucial to clarify which blood cells are expressing HMMR and being quantified. This information would strengthen the interpretation of the data.
RESPONSE: We have modified the methodology and results to make the quantification of HMMR by quantitative RT‒PCR and immunohistochemistry understandable.
COMMENT: The authors should also propose potential approaches for targeting HMMR therapeutically and outline how these could yield translational insights for clinical applications.
RESPONSE: This observation is very valuable; however, further studies are needed to identify HMMR as a therapeutic target. HMMR is not an ideal therapeutic target [1, 2]. However, we also think that there are coexpressed variants in leukemia that could have opposite functions. However, to do so, it is necessary to be able to determine whether the variants are the cause of HMMR being expressed in different areas of the cell. The most important clinical application at the moment is as a marker that helps us predict early which patients are at high risk of death.
- Snauwaert, S., et al., RHAMM/HMMR (CD168) is not an ideal target antigen for immunotherapy of acute myeloid leukemia. Haematologica, 2012. 97(10): p. 1539-47.
- Schendel, D.J., Is it time to abandon RHAMM/HMMR as a candidate antigen for immunotherapy of acute myeloid leukemia? Haematologica, 2012. 97(10): p. 1454-5.
COMMENT: Additionally, the manuscript should address differences in HMMR expression across various types of leukemia and other hematological malignancies, as this is necessary for inclusion in the Discussion section.
RESPONSE: We have added information on this point both in the introduction and in the discussion, since it is very important to note that HMMR is expressed in different types of leukemia; however, B-ALL is the most important for us because it is the most frequent in our country and for our population.
COMMENT: The authors should provide an explanation for the lack of statistically significant differences in HMMR expression between relapsed and non-relapsed B-ALL patients, as this discrepancy appears to be unrelated to the amount of HMMR expressed.
RESPONSE: We have added information on the discrepancy in expression between relapse and death in B-ALL patients.
COMMENT: Finally, in Figure 5B, the Y-axis label is missing and should be added for clarity.
RESPONSE: Figures including Figure 5B have been corrected.
Round 2
Reviewer 1 Report
Comments and Suggestions for Authors
The authors added more data points in microarray and showed specific dilution factor for their cytospin data. I suggest accepting this revised manuscript for publication.
Reviewer 2 Report
Comments and Suggestions for Authors
The authors have already answered the questions and improved their manuscript.